# Periurban Transformations in the Global South and Their Impact on Water-Based Livelihoods

**Carsten Butsch * and Sophie-Bo Heinkel** 

Institute for Geography, University of Cologne, Albertus-Magnus-Platz, DE-50923 Cologne, Germany;
s.heinkel@uni-koeln.de
* Correspondence: butschc@uni-koeln.de; Tel.: +49-221-470-4142

**Abstract:** Urban sprawl and population increase are fundamentally transforming periurban areas in the Global South. These areas often suffer from inadequate environmental planning, resulting in water sources being overexploited, degraded, and redistributed. These processes affect water-based livelihoods due to disadvantages in water access and inadequate water governance. On the positive side, these transformation processes are leading to alternative water-based livelihoods. We systematically review and critically comment on the literature on water-based livelihoods in periurban areas of the Global South to provide the current scientific knowledge on this topic. Transformations of water-based livelihoods in periurban areas were also evaluated in terms of their sustainability. We conclude that rapid developments of periurban areas contain threats and potentials for water-based livelihoods and some emerging water-based livelihoods, whereas some emerging water-based livelihoods provide interim solutions for institutional supply gaps. Major lacunae in research are the (1) lack of holistic approaches, which address social dimensions of transformations, (2) the lack of studies applying a differentiated perspective on neighbouring areas within the urban fringe and (3) a lack of knowledge on emerging (water-based) livelihoods.

**Keywords:** periurban; Global South; water; livelihoods; agriculture

## 1. Introduction

Periurban spaces in low and middle income countries are zones in transition between the city proper and rural spaces [1]. Their transitional nature has three dimensions: first, they are the zone between the urbanised, built up area and the rural landscape. The land use changes in this zone with a tendency of more intensive land use towards the core and less intensive land use towards the periphery. Second, many periurban spaces are experiencing transitions from one land use to another. Third, periurban spaces are prone to governance transitions, with established institutions disappearing and new actors appearing, resulting in an increase of informal decision-making processes and developments. The transitional nature of periurban spaces is both a threat and an opportunity for sustainable development—of these spaces themselves but also the larger metropolitan areas they are located in.

Periurban spaces are rural-urban interfaces – formerly rural spaces transformed, because of the growth of cities, relying on the resources of their surroundings. While the term "periurban" is also used in other regions of the world [2–7], this review focuses on the Global South. Here, the close connection to urban centres brings capital and ideas to periurban spaces and creates new markets for periurban entrepreneurs [1,8–13]. Thus, periurban spaces are spaces of flows, where the rural and the urban meet and exchange and that are characterized by a mix of urban and rural features, resulting in a mosaic of land uses [14,15], a multiplicity of stakeholders—sometimes with diverging interests [16]—and in many cases weak governance structures [1,11,12]. The flows of people consist of

commuters, (temporary) migrants to the city, and high income groups moving out of the core city; goods are exchanged in both directions, with the periurban delivering agricultural and industrial products to the city, while the city provides consumer goods and access to global markets; flows of information and capital usually are unidirectional from the city to the periurban spaces; so are flows of waste, in which periurban areas are used as sinks—though wastes might also become valuable resources (Figure 1).

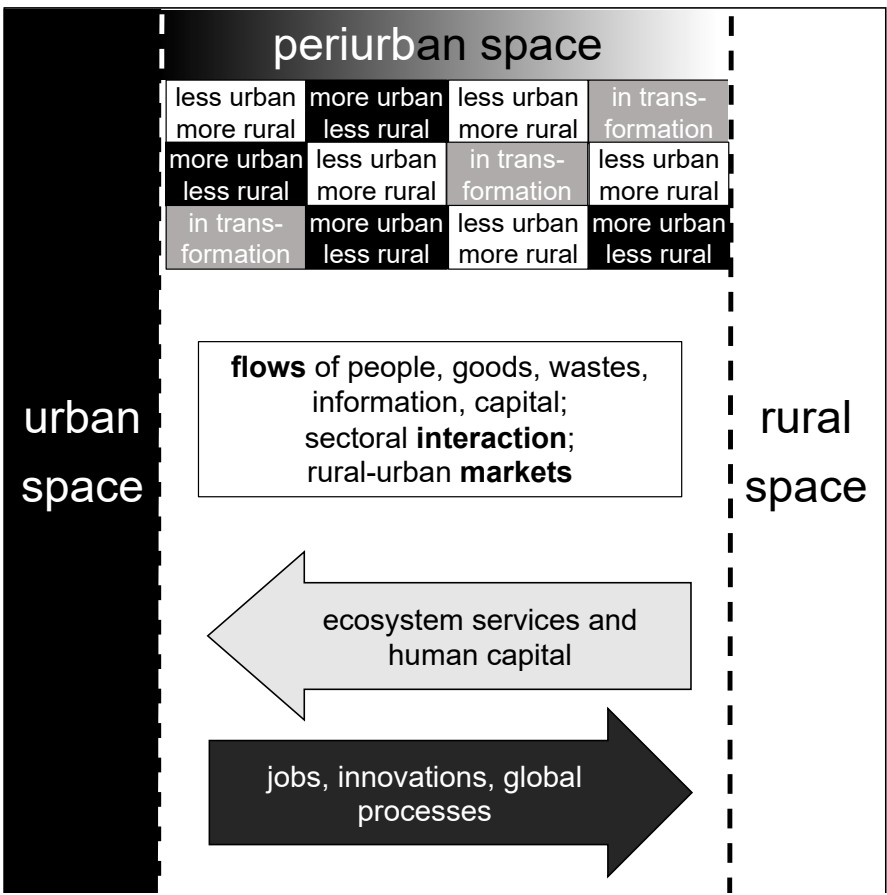

**Figure 1.** The periurban as a space in between.

The directions of these flows put water as a resource in periurban spaces under pressure: while water demand is increasing in situ and in the core city, simultaneously the resource is being depleted. The internal demand is related to the intensification of water-based activities in the periurban, e.g., increasing agricultural or industrial production, and the increase in population. At the same time, the growing cities demand more water, which in many cases has to be sourced from periurban areas. In addition, periurban water bodies are used to dispose industrial effluents and sewerage from the urban areas, with the ensuing degradation of both the surface and the ground water quality [10,17–19].

Together with overextraction and pollution, changes in managing the resource, result in changing access to water, with the patterns being determined inter alia by location, socio-economic status, gender, and social identity [18,20–24]. In many cases, the shortfall of governance is the root cause for these problems. Often, institutions that were initiated to govern villages have to handle urbanisation processes. At the same time, institutions which traditionally managed common goods erode and new actors, e.g., real estate entrepreneurs, enter the scene. Thus, traditional water management ceases to exist without new forms being established [25–27].

This review portrays how water-based livelihoods in periurban areas of the Global South are affected by multidimensional transformation processes, affecting the physical space (transformation of

land use, depletion of aquifers, changes in the local climate etc.) and the society (population structures, changing occupation patterns, governance etc.). This review aims at providing an overview of and critically commenting on different strands of research that addresses water-based livelihoods. In doing so we will identify lacunae in the existing literature and develop an own research agenda to tackle these.

This work is part of the project "H$_2$0-T2S—in urban fringe areas" funded under Belmont Forum's and NORFACE's joint program "Transformation to Sustainability". The project aims at understanding how transformation processes in urban fringe areas of Indian metropolises influence the access to water as a consumption good and a resource for livelihoods. There are spatial, social and economic aspects that need to be understood given the fast-changing land-uses and employment shifts in these areas. These include reduced or discontinued access to water for livelihoods due to overextraction or pollution, gender- and cast-specific livelihood patterns cross-cutting with social identities and the location specific characteristics of these aspects [21,23]. Going beyond an analytical approach the project will in the second phase develop alternative, more sustainable transformation pathways with local stakeholders.

## 2. Materials and Methods

The two authors conducted a systematic literature review, using the PRISMA-guidelines [28,29] as an orientation. By screening the databases Scopus and Web of science with a defined set of keywords as listed in Table 1, we systematically reviewed 517 articles published in peer-reviewed journals between January 1991 and January 2019. Of these titles and abstracts were screened with the Covidence systematic review software (Veritas Health Innovation, Melbourne, Australia, available at www.covidence.org). All articles considering periurban settings in the Global South were included in the second step of the literature review; articles addressing either explicitly urban (e.g., urban agriculture) or rural settings were excluded. Also, technical and clinical studies were excluded, as were articles focusing on Western countries only. Conflicting votes were resolved by a joint screening of the abstract. The remaining 125 articles were subject to an independent full text screening by the authors. During this process, core aspects of the article were tabulated. On this basis a final decision regarding the inclusion of the article was taken. Diverging votes were resolved in discussion. This systematic text screening resulted in the inclusion of 53 articles on water-based livelihoods in periurban areas, from which data was extracted (Figure 2) for content analysis.

Both databases, Scopus and Web of Knowledge, provided a high degree of congruence for the same keyword patterns. Additionally, different keyword patterns often provided the same articles, both explaining the high number of duplicates in the initial database (614) (Table 1). This high number of duplicates indicates, that both databases cover a similar body of literature and that the search patterns applied cover a large body of the scientific literature on the topic.

In our analysis we found that different aspects of water based periurban livelihoods have been covered very differently in the literature. On the one hand the development of periurban agriculture (production of vegetables and crops, animal husbandry) appears to be very well investigated, while on the other hand no sources were found, which addressed livelihoods related to materials such as textiles and soils in making explicit reference to their situatedness in periurban areas (tannery, brick making, pottery, etc.). These types of livelihoods seemed to be neither issued in scientific literature, nor properly investigated. This lack of knowledge on these topics is one of the first lacunae identified through our review, especially as these activities are potentially threaten sustainable development pathways. These livelihoods can heavily affect the local ecosystem, as they can harm or destroy soils if they are extracted (brick making, concrete making) or polluted (industrial effluents from informal factories).

The main body of literature (70%) included in the qualitative synthesis is older than five years which bears the danger of missing out on the most recent debates on livelihoods in periurban areas. Thus, we extended our literature research pattern for not-peer-reviewed documents published after 2015 and included 24 additional sources after full text screening. The temporal distribution of the

articles included shows that periurban livelihoods are topic that has received growing attention since the mid of the last decade, with the number of publications being relatively stable over the last ten years (for 2019, only publications up to June were included in this review, so the figures for that year are not representative) (Figure 3).

**Table 1.** Keywords used for identifying relevant literature.

| Date | Literature Database | Search Query | | |
|---|---|---|---|---|
| 23 January 19 | Scopus | water | AND | AND |
| 24 January 19 | Web of Science | | "periurban" | agriculture |
| | | | periurban | "animal husbandry" |
| | | | "urban fringe" | "brick mak *" |
| | | | | "cattle keep *" |
| | | | | "clean * of clothes" |
| | | | | "concrete block mak *" |
| | | | | farm * |
| | | | | fish * |
| | | | | "fish farm *" |
| | | | | "food vend *" |
| | | | | horticulture |
| | | | | laundry |
| | | | | livelihood * |
| | | | | "livestock farm *" |
| | | | | pottery |
| | | | | poultry |
| | | | | tan * |
| | | | | "textile produc *" |
| | | | | "water vend *" |

* = Wildcard character: Search operator, which matches any number of potential additional characters of the root of a word.

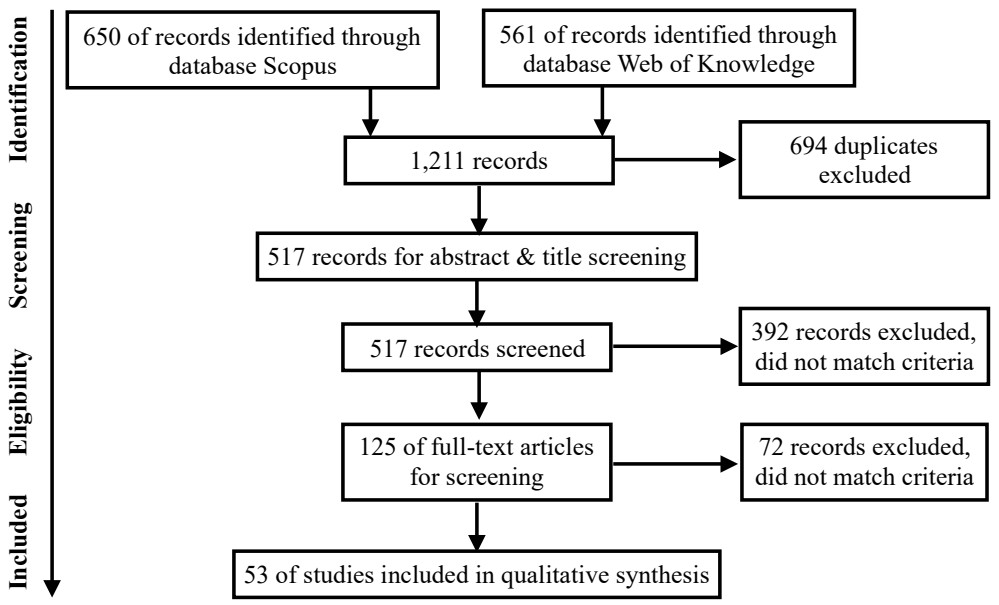

**Figure 2.** Process of the systematic review.

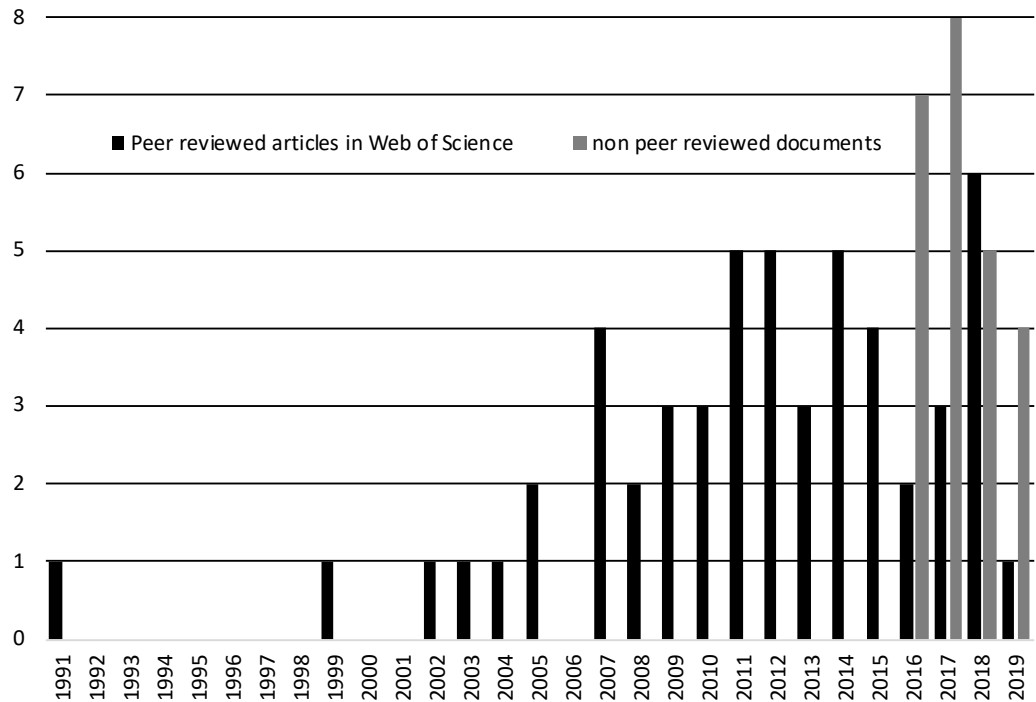

**Figure 3.** Temporal distribution of publications reviewed.

## 3. Results

Three thematic clusters were identified in the literature on water based periurban livelihoods: (1) agricultural responses to resource scarcity, including wastewater use, (2) aquaculture and fishing, (3) emerging water-based livelihoods. There is a lack in literature applying an original sustainability perspective, which is our field of inquiry. Thus, the findings described in this section will be discussed regarding their meaning for sustainable transformation in section four.

A difficulty, we faced in the analysis is that in many cases, it is hard to differentiate between water use for domestic purposes and for income generation, as households are consumers and small scale producers of agricultural products or service providers who need water as a basis for their livelihoods [30]. A common theme across the literature on different livelihoods was the decrease of access for traditional users inter alia because changing governance structures: former commonly managed goods (ponds, lakes, rivers) are now managed by governments, which limit access by issuing leases [31] or claim earlier sources of irrigation water to supply drinking water to urban areas [32].

### 3.1. Agriculture

In periurban areas, agriculture battles with other land uses over scarce resources such as water and labour, with pressure usually increasing during the transformation process. A more general perspective on agriculture [33] states that "urbanisation can create both, threats and opportunities for peri-urban farms". Water related risks also increase, e.g., losing harvest completely due to water shortage or pollution [34–37].

### 3.1.1. Different Strategies to Cope with Changing Access

Farmers differ in their ability to cope with the increasing pressure on water, land, and labour. They must adapt their production practices to the permanently changing surrounding, often shifting away from subsistence farming to market-oriented production. The agricultural practices of farmers depend on their market access and the availability of inputs. More rural areas—which can also be in close proximity to the city (cf. Figure 1) – are described to follow more subsistence oriented forms of agriculture, while urbanised areas show an intensification towards commercial farming [38,39].

Depending on how land, labour, and capital are recombined, different agricultural strategies develop, namely expansion, shrinking, intensification, de-intensification [33,40,41]. Alternatively, a diversification of livelihoods can be pursued, when household seek additional non-agricultural income sources (see Section 3.3). One potential strategy in this context is temporary migration. The overall picture emerging from the literature is however, that farmers have very limited power in transformation processes and thus very limited choices.

Marginal farmers lack the financial means and/or secured permanent ownership of land to follow an expansion or intensification strategy [33,42]. Their possible alternative production strategies require more time because their adaption capacities are weaker [43]. Options for these farmers are the diversification of income or to experiment with new crops [33,44], e.g., by shifting to floriculture or market-orientated cultivation of vegetables [45]. A short fall of the literature in this context is the absence of studies addressing the options and strategies of those who are driven out of agriculture by the transformation processes. Especially landless labourers, who are not compensated during the transformation process (as the land-owning farmers are), are potential losers of livelihood transformations in periurban areas—though the literature largely remains silent about them.

Only farmers with a solid asset base can manage transformations in a self-determined way. In some regions, where the water table is depleting fast, farmers face the option to 'go deep or quit' [46,47], i.e., to invest in deeper borewells or seek alternative occupations. Failure in water governance result in worse access to water for female farmers [42]. Especially marginal farmers have to quit who lack the financial means to invest in deep boreholes [24,47,48]. Farmers losing access to their traditional water sources also leave their land temporarily fallow, buy water from outside, shift to wastewater, or move out of agriculture [47] and may migrate permanently to the city [41,49,50]. Sometimes, male family members seek alternative occupations an commute to the urban centre on a daily basis, resulting in increasing responsibilities of females on farms [50,51]. These changes in livelihoods eventually result in deep societal transformations of villages, which have not been studied holistically.

Wealthier farmers have more options to become winners of transformation processes. They can respond by intensifying their production as the demand for fresh vegetables, meat and dairy products by the growing urban population increases while the available land for agriculture decreases [42,45,46,52–54]. In this way they can generate additional income for their households. However, even better-off farmers face problems in securing access to water sources and may eventually have to diversify their water sources [36,39,45,53] and adjust their production patterns to the incremental loss of open space and grazing areas [55]. Wealthier farmers may respond to emerging water conflicts through periurban transformations by moving their farmland location or farming on previously unused land [56]. This can result in a cut-throat competition for arable land in peripheral areas of the urban fringe.

These current developments in periurban agriculture are mainly judged as unsustainable transformations [42,46]. A transformation to sustainability needs to take into account the multiple tasks periurban agriculture can fulfil, including its contribution to food security, energy delivery and the preservation of natural resources and the provision of ecosystem services [31,33,38,56]. Periurban governance has to take into account the complexity of periurban structures, actor constellations and processes [40,47]. Several authors demand the creation of efficient water markets that ensure equitable distribution, improved accessibility, eliminate common good problems and mechanisms that may eventually compensate farmers for the delivery of ecosystem services [20,32,51,57,58]. What is missing is research on the societal impact of agricultural transformations. Agriculture as a livelihood is strongly influencing social interactions and the general organisation of life. One aspect are the multiple connections between land owning farers and landless agricultural labourers. Periurban transformations change these relations fundamentally, yet this has so far not adequately captured in the literature.

### 3.1.2. Wastewater Irrigation for Agriculture

Within the cluster on periurban agriculture, a sub-cluster addresses issues of wastewater irrigation. The largest share of that literature describes positive effects of farmers shifting to wastewater irrigation: (1) it allows for expanding to previously barren land [59]; (2) farmers can grow crops with less or even no fertilizer while simultaneously closing nutrition cycles between urban and periurban regions, as the nutrition included in the wastewater had been imported to the city earlier [60–64]; (3) it is in many cases cheaper, though not necessarily free of cost, sometimes farmers even invest in infrastructure to get access to the resource [65,66]; (4) in several studies farmers described a yield increase [52,60,63]; (5) wastewater is constantly available round the year, which may not be the case for surface water or even ground water [52,60,63–65,67,68]; (6) it offers higher returns for farmers who aim at producing middle or lower quality products [67].

The potential negative health effects resulting from the use of wastewater are discussed by several authors [62,63,69,70]. Several studies analyse the chemical and microbiological contamination of wastewater used for irrigation [71–79]. Main studies on microbiological contamination are on total coliforms, Escherichia coli and Salmonella, which are related to fecal contamination [73,77,78,80,81]. The contamination on agricultural products poses health risks and often exceeds the WHO guidelines for unrestricted irrigation. In the case of Hanoi the load with Salmonella was 260-fold the tolerable safety limit [78]. Chemical pollution is related to heavy metals from industrial effluents and fertilizers as well as soil parent materials [71,74].

Farmers using wastewater have a significantly higher risk of gastroenteric diseases and intoxication [78,82], while others highlight the possibility of safe use of wastewater, if it follows prescribed, e.g., the WHO guidelines on the safe use of wastewater, excreta and greywater [69,83]. In some areas wastewater irrigation had an impact on crop quality resulting in a reduction in market prices, inter alia due to the consumers' concerns about potential health risks from these vegetables [56,64,67,70]. Regarding this wastewater irrigation is also seen critically, and only picked up by marginal farmers who lack alternatives [84].

Wastewater irrigation can be part of strategies towards more sustainable regional development and as a possibility to improve farmers' livelihoods [60,68,69]. Two main arguments are named: (1) the use of wastewater allows for closing nutrition cycles as cities provide nutrition for periurban farming, while agriculture provides food for the cities serving at the same time as wastewater purification zones [61,66]; (2) it can contribute to poverty alleviation, if farmers get the financial means to expand their business, e.g., through micro financing [77,85] and if land tenure is secured [36].

On top of this, wastewater is a potential sustainable alternative to depleting fresh- and ground-water resources [63,86], which would be the preferred sources of irrigation. Yet the quality of the wastewater determines which crops or flowers can be grown successfully [62,72]. If sewerage from households is mixed with industrial effluents, wastewater becomes unsuitable for irrigation. Polluting industries, emerging as new—often also water-based – livelihoods in periurban areas, can thus destroy traditional water-based livelihoods. Thus, only wastewater fulfilling certain criteria is suitable for irrigation, which requires adequate management of this potential resource [45,65,68].

Wastewater management in periurban areas in the global South urgently needs to be improved in order to leverage its potential for regional sustainability strategies. It is imperative to eliminate health concerns related to wastewater irrigation. This includes the separation of rainwater, wastewater from households and industrial wastewater. There is a need for multi-stage purification for different types of wastewater. Only selected types of wastewater or, in the best case, pre-treated wastewater, such as retention filtered, should be used for agricultural purposes. Another aspect, that has so far not been researched adequately is the societal dimension of waste water irrigation, which not only has a technical dimension. It is for example related to societal/cultural/ritual aspects of purity (e.g., in India), which is often overlooked by authors focussing exclusively on measuring the nutritional value of wastewater.

### 3.2. Aquaculture

The second, smaller cluster of literature concentrates on aquaculture and fishing as an emerging livelihood in periurban areas. Aquaculture provides ecosystem services, such as cleaning of wastewater, with the East Kolkata Wetland being one of the most prominent examples [70,87].These wetlands receive about one third of Kolkata's effluents, which are here the nutritional basis for intensive fish-farming. Here and elsewhere wastewater-fed aquaculture permits the nutrition cycles between periurban and urban spaces to be closed [87–89].

Aquaculture is a means of intensification or specialisation that can increase farmers' incomes. Nevertheless, a specialisation in aquaculture or fishing may also result in increasing vulnerability, if it means a reduction of livelihood options [90]. Additionally, aquaculture as a follow-up use for erstwhile agriculturally used fields reflects a means of intensification, albeit one requiring large initial investments only few farmers can afford [57,87]. Problems are related to the overuse of resources and the eutrophication of soils and water [44]. Monocultures of certain fish species are common, however the species vary between different regions [91]. Wastewater-fed aquaculture can be prone to heavy metal and microbial contamination, which affects the fish breeding in these systems, posing health risk for farmers and consumers [78,92]. Though two issues of this periurban transformation are not adequately addressed: First, aquaculture as an intensive land use is destroying the soils, which means that a transformation to other land uses than built up area is afterwards impossible. Thus, the decision for aquaculture is already predetermining the direction of development. Second, in aquaculture there often is an overuse of antibiotics. The release of antibiotics in densely settled areas and especially in connection with wastewater can create harmful resistances.

Fishing in natural water bodies is a traditional livelihood, often relevant for socially deprived communities. As a traditional livelihood it is facing increasing competition in periurban areas for the scarce resource water. Access to water bodies traditionally used for fishing is renegotiated, often resulting in a loss of access for traditional fishing communities, which in many cases are located at the lower end of the socioeconomic spectrum [31,90]. The question, whether fishing could be part of an integrated landscape management has so far not been researched in detail, but seems worth further consideration.

Sustainability issues addressed in the literature on fishing are related to governance, mainly the lack of access to water bodies for deprived communities [51,90]. The literature on aquaculture, ecosystem services [87] and the chance to close nutrition cycles are main issues discussed [87–89]. Recently, new ways to make aquaculture more eco-friendly, include deprived communities and thus contribute to sustainable growth have been discussed [93,94]. In Nigeria, for example, it is planned to establish an eco-friendly aquaculture system in mangroves forests which is based on polyculture and is adapted to the mangrove ecosystem. Therefore indigenous knowledge is utilised while simultaneously observing local people's priorities [93,94]. Projects like this show the potential for sustainable periurban transformations, while currently aquaculture often results in unsustainable development.

### 3.3. Emerging and Diversification Water-Based Livelihoods

Water vending received most attention so far among the different emerging water based livelihoods, which form the third cluster, identified in the literature review. In the sparse literature a diversity of water-based livelihoods is described [30], e.g., polluting small-scale industries like dyeing [95], brick making [96], food vending [97] or tourism [98]. In many cases, industrial production and services require water, often polluting it such that it is no longer feasible for irrigation.

Water vending and the emergence of a private water market can be seen as an expression of planning failures. Although water vending is a traditional occupation in some regions of the world, it has increased in the last decades, mainly in places where traditional water sources have become unavailable and existing utilities do not work according to prescribed norms, making people more willing to pay for water [99]. Water vendors in some places cater to urban populations, in some places to periurban populations. In either case, the new small water enterprises bridge gaps in the existing

infrastructure that result from rapid urban growth [97,100] and are viewed positively by economists who highlight, that they work without subsidies [101]. Their contribution to the stabilisation of water supply has so far not been adequately captured in water frameworks [102]. Yet in many cases this water supply comes at very high cost—up to 40 times the price consumers with a piped connection pay per litre – for the consumers, for whom water vendors may be the only source of water [102].Thus, water vending has to be assessed ambivalently: In the current situation it is the best response to governance failures, in the long run, proper water connections for each household are needed in periurban areas.

For well off farmers, water vending is a chance to increase their income, while it deprives poor farmers who cannot afford borewells from their livelihood base. Water vending generally requires less labour input and offers higher profits [20,22,103]. Yet, this new livelihood results in further transformations, as farmers drop out of agriculture and professionalise in water vending [20,22]. This is only possible for well-off, land owning farmers, who have the means to invest in drilling deeper boreholes and water tankers [24] and/or treatment facilities such as reverse osmosis plants. This moving out of agriculture by the land-owners leaves landless labourers previously engaged in agriculture without a livelihood [24] and results in a decline of the water table [20]. Thus, the transfer of water from the periurban to the urban areas may also result in a drinking water scarcity in periurban areas [22].

Emerging livelihoods compete with traditional livelihoods—contributing to a shift of occupations [20,103]. Yet, they are also causing overexploitation of resources and a depletion of water quality [24,95]. Positive aspects are related to the contribution of water vendors to increase the access to water and to the creation of new non-agricultural forms of income [102]. Yet, these aspects have hardly been researched so far. As this review illustrates, emerging water-based livelihoods other than water vending have not been captured adequately in the literature. This is a severe research gap that needs urgent attention, as they are essential for understanding the ongoing periurban transformation.

*3.4. Spatial Aspects of Water-Related Livelihoods*

The periurban in-betweenness and the relative location to the city fundamentally shape access to water for livelihoods. Communities located upstream of cities experience a decrease of fresh water quantity due to the emerging and exploiting flows of water towards the "thirsty cities" [34,49]. The lack of knowledge about the complexity of the local hydrological system makes it easy for stakeholders from urban spaces to use their power and extract fresh water from the upstream communities without adequate compensation [34,49]. In contrast, the water quantity of communities located downstream of cities increases through wastewater discharges, which also reduces the water quality [34]. The availability of surface water and the provision of wastewater treatment gives communities downstream under certain circumstances (depending on the legal framework) an advantaged position in negotiations about water rights with water authorities [49]. The downstream position can also be disadvantageous if wastewater from the city is contaminated and therefore not suitable for, e.g., agricultural use, and at the same time institutions lack negotiation power. Thus, the upstream-downstream power constellations depend on the governance framework in place. The linkages between peri-urban areas and the power constellations on the regional level are decisive for sustainable development, as formulated in SDG target 11.A. These linkages need further attention, especially the spatial aspect of access to water for livelihood generation needs further investigation.

## 4. Discussion

The governance settings in periurban areas of the Global South are not adequate to steer the transformations observed. Fundamental changes of the physical landscape and societal structures take place in an institutional quasi-vacuum with old institutions like village councils losing their power and new institutions not being in power, financially potent actors shape these transformations. These profit-oriented actors often steer development into unsustainable directions, as they overextract or pollute water, limit the access of other groups or distract economically weaker strata of the society from their traditional water-based livelihoods.

Often, people and institutions in periurban areas do not have the power to negotiate their own interests. An independent development of periurban spaces as spaces in their own right hardly is hardly possible under these conditions. Institutional actors like municipal governments for example restrict the development options of periurban spaces when they degrade periurban spaces to loci of resource extraction for the "thirsty city" [49] by playing out their institutional power.

At times, temporary unsustainability seems unavoidable during the transformation. Water vendors are in some cases the only source of drinking water for periurban communities, securing basic needs [102]. They deliver services in an institutional vacuum, where the traditional rural water supply, based on wells, is no longer available and new forms of water supply have not yet emerged. Thus, any assessment of practices like water vending has to differentiate between a distanced resource extraction for the city, hindering development, and a localized resource extraction for the periurban areas, which can at least temporarily be required to fulfil sustainability goals like universal access to water. Further research on water vending could provide knowledge on coping strategies of water-vulnerable and the systematic analysis of water vending networks could provide knowledge on spaces of water vulnerability in periurban areas.

On the long run, new planning mechanisms are needed that respond to the complexity of periurban spaces and that allow for different outcomes (Figure 4). Often periurban spaces are conceptualized as spaces with predetermined transformation pathways resulting in urbanisation ("cities to be" [104]). Yet, they can follow individual trajectories and could also fulfil important functions within metropolitan regions, e.g., provision of ecosystem services. Regarding sustainable water-based livelihoods transformation pathways have to consider a high degree of just accessibility to water, ensuring a protection of the resource. Planning processes need to take into consideration the needs of people in periurban spaces and must not take place from an urban perspective. Periurban spaces need the chance for a development in their own right and not only as urban appendices. For this, a more systematic approach towards periurban spaces is needed. The development trajectories of periurban spaces (Figure 4) need to be analyzed with the goal to develop a typology of areas within periurban spaces.

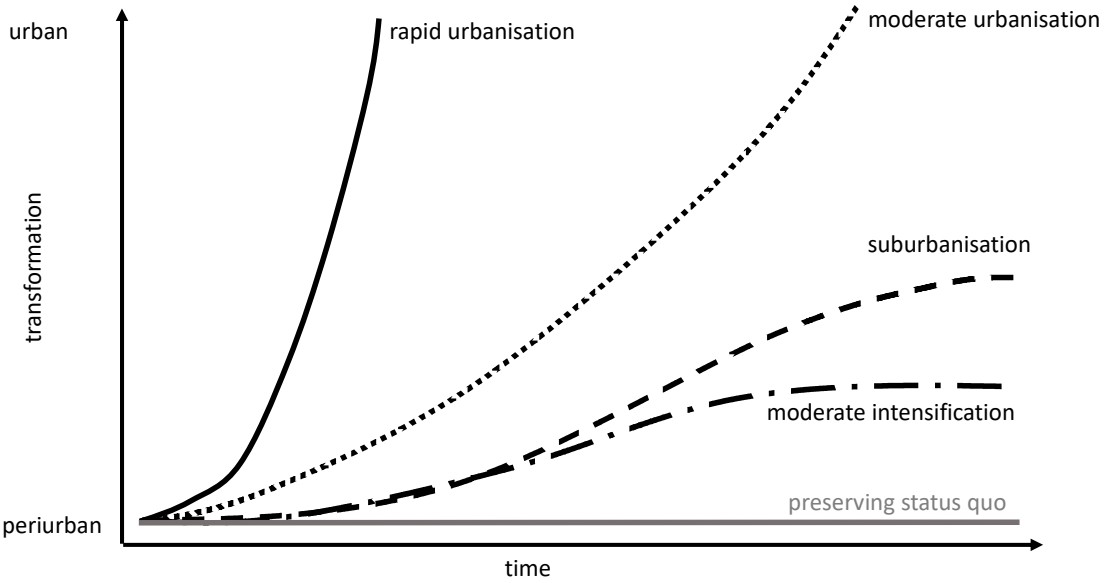

**Figure 4.** Development trajectories of urban fringe areas.

Several starting points for the sustainable development of periurban areas have been identified in the literature with relation to water and water-based livelihoods. These have not been transferred into consistent strategies towards sustainable transformations yet. The development of regional systems to deal with wastewater offers a high potential. These regional waste-water management systems have to

allow for leveraging the benefits related to wastewater irrigation (closing of nutrition cycles, availability around the year, purification by retention and filtering functions of soils) and simultaneously eliminate the health threats for producers and consumers. More sustainable transformations could also be achieved by the creation of water markets, which—if smartly shaped – help to avoid misallocation of the scarce resource. Especially restructuring periurban agriculture can result in a more robust periurban primary sector, which can fulfil several tasks, inter alia poverty alleviation, food security, provision of ecosystem services. This restructuring also includes measures like payment for ecosystem services. To implement these ideas new and stronger governance frameworks are needed, which are robust in times of transformation and adapted to periurban complexity.

Several lacunae were identified in the existing literature, which translate into an agenda for further research on the inevitable transformation of water-based livelihoods in periurban. From our point of view, research on periurban transformations needs to address three major issues: First, the social dimension of the periurban transformation in general and of water-based livelihoods in periurban areas specifically has not been addressed adequately. The dissolution of traditional village structures with reciprocal connections between different occupation groups needs to be investigated. Second, emerging (water-based) livelihoods have to be identified and analyzed. Knowledge about small informal industries, day tourism, services etc. and their impact on the socio-ecological systems is sparse, but urgently needed to steer development processes into more sustainable directions. Third, research on livelihood transformations in periurban areas needs a more solid theoretical basis. New perspectives within political ecology [105], approaches like the livelihood landscape [106] or a spatial perspective on changing access [107] is needed to address issues like changes of power relations, interactions, conflicts and relationships of residents with new population groups and changing gender relations.

Overall, research needs to investigate good practice examples to identify success factors for transformations to sustainability. In participatory processes, potential transformation pathways need to be developed to secure open-ended future development options (Figure 3).

## 5. Conclusions

Which pathways are followed in urban fringe areas—as specific spaces of flows—will be decisive for the overall sustainability of the growing metropolises of the Global South. They will then either become a of the metropolis itself or will continue to provide ecosystem services, simultaneously drawing capital and investment from the city. Either way, the transformation of periurban and urban spaces is entangled in multiple ways.

The described changes in water-based livelihoods are an expression of this entanglement. In most periurban areas, established water-based livelihoods are lost or changed while new ones emerge. Some of these emerging livelihoods are temporary in nature as they fill current gaps in governance, e.g., water vending, but might become obsolete once the transformation to an urban space are completed. Others offer solutions towards sustainable and localised production patterns, if properly managed, e.g., wastewater-fed agriculture and aquaculture.

In many cases, old and new water-based livelihoods are competing over the increasingly scarce resource. Inclusive, community-based planning process are needed to develop sustainable transformation pathways. These should (a) reflect the complexity of periurban spaces and (b) leverage the potential of water-based livelihoods to simultaneously provide ecosystem services and contribute to poverty alleviation.

Based on the analysis of literature we find three major points for a future research agenda on water-based livelihoods: First, research needs to address the social aspects of transformation, second, more knowledge is needed on emerging water-based livelihoods and third, research needs a stronger theoretical foundation. Research on the urban fringe in general has to treat periurban areas in a more differentiated way. The different periurban development trajectories need to be systemized to be able to develop sustainable development pathways, at best with participatory methods.

**Author Contributions:** Conceptualization, C.B. and S.-B.H.; methodology, C.B. and S.-B.H.; formal analysis, C.B. and S.-B.H.; data curation, S.-B.H.; writing—original draft preparation, C.B. and S.-B.H.; writing—review and editing, C.B.; visualization, C.B.; project administration, C.B.; funding acquisition, C.B. All authors have read and agreed to the published version of the manuscript.

**Funding:** The project H2OT2S is financially supported by the Belmont Forum and NORFACE Joint Research Programme on Transformations to Sustainability, which is co-funded by DLR/BMBF and the European Commission through Horizon 2020 under grant agreement No 730211. Within this programme, this research was funded by the German Federal Ministry of Education and Research (BMBF) under the grant number 01UV1814.

**Conflicts of Interest:** The authors declare no conflict of interest.

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
