# Peer review of "Periurban Transformations in the Global South and Their Impact on Water-Based Livelihoods"

_water, doi:10.3390/w12020458_

Round 1

Reviewer 1 Report

Dear Authors, kindly add some more year-wise results regarding periurban water-based studies from reference work in graphical /tabulator/geographically form for the betterment of review article and check English sentence structure, Grammar error, and Figures mistakes throughout the manuscript. Also, reviewed and modified the manuscript as per comments given below:

Line 18-19 The statement should be revised as “water-based livelihoods, whereas some emerging water-based livelihoods provide interim solutions for institutional supply gaps”.       Line 19-21, English error, author should be revised the statement in logical way. Line 27, what is meaning of word “city proper” in the statement. Line 71-72, Authors should replace the sentence regarding to define “Periurban” after Line 37. The Table title should be above the concerned table check throughout the manuscript. Line 89, Replace Web of Knowledge to Web of Science. Line 90, Table & Figure word should be uppercase in the text body. Line 107, Table 1 should be formatted as per Water journal guidelines. Line 115-116, The sentence should be spilt in two statement with a proper expressing contrasts. Line 107, write the * in Table 1, define in header of Table. Line 145, Author suggested to add some more results regarding to Agriculture section with some updated citation. Line 416, Figure 3. Should be revised to Development to potential trajectories of urban fringe areas Line 417-441. Author suggested to concise the Conclusion section in a technical way. and delete statements have similar meaning.  

Author Response

Comment 1: kindly add some more year-wise results regarding periurban water-based studies from reference work in graphical/tabulator/geographically form for the betterment of review article

Response: A graph was inserted showing the temporal distribution of articles.

Comment 2: Line 18-19 The statement should be revised as “water-based livelihoods, whereas some emerging water-based livelihoods provide interim solutions for institutional supply gaps”

Response: Thank you very much for this comment, we have changed the sentence accordingly.

Comment 3: Line 19-21, English error, author should be revised the statement in logical way.

Response: Thank you very much, we revised this accordingly

Comment 4: Line 27, what is meaning of word “city proper” in the statement.

Response: “city proper” refers to the city in its administrative boundaries. Please refer inter alia to the UN data booklet “World cities 2018” (https://www.un.org/en/events/citiesday/assets/pdf/the_worlds_cities_in_2018_data_booklet.pdf), page 1: “One type of definition, sometimes referred to as the“city proper”, describes a city according to an administrative boundary.”

Comment 4: Line 71-72, Authors should replace the sentence regarding to define “Periurban” after Line 37.

Response: Yes, thank you, we inserted the sentence after line 37.

Comment 5: The Table title should be above the concerned table check throughout the manuscript.

Response: Yes changed this.

Comment 6:Line 89, Replace Web of Knowledge to Web of Science.

Response: Thank you very much, changed this accordingly.

Comment 7: Line 90, Table & Figure word should be uppercase in the text body.

Response: Changed this in the whole text accordingly

Comment 8: Line 107, Table 1 should be formatted as per Water journal guidelines.

Response: Yes it was reformatted

Comment 9: Line 115-116, The sentence should be spilt in two statement with a proper expressing contrasts.

Response: We rephrased the sentence, hoping, that it’s clearer now.

Comment 10: Line 107, write the * in Table 1, define in header of Table.

Response: Yes, this was changed accordingly.

Comment 11: Line 145, Author suggested to add some more results regarding to Agriculture section with some updated citation.

Response: The whole section 3.1 is on agriculture and the references 33-58 are the ones we found applying the PRISMA method, as suggested by the journal editorial board. These sources are covering a period from 2003 to 20019, with the majority is more recent. We are quite uncertain, how to deal with this comment. Maybe the reviewer could be a bit more precise on how we could add literature without compromising on our methodology and especially by giving a hint on which further resources to include.

Comment 12: Line 416, Figure 3. Should be revised to Development to potential trajectories of urban fringe areas

Response: Thanks for this suggestion, but this title would not get what we would like to express. Thus we changed the title to “Development trajectories of urban fringe areas”, which is hopefully clearer.

Comment 13: Line 417-441. Author suggested to concise the Conclusion section in a technical way. and delete statements have similar meaning.

Response: Thank you, we tightened the conclusion, hoping that it is acceptable now.

Reviewer 2 Report

The paper by Carsten Butsch and Sophie-Bo Heinkel with title ”Periurban transformations in the Global South and their impact on water-based livelihoods “ on a subject which is acknowledged to be difficult, is important for several reasons.

The authors conducted biblical research useful for the perspective of their research project.

The research conducted is not exhaustive.

Data processing is not of general scientific interest.

The conclusions have no impact on scientific progress.

Future research results will be interesting.

Author Response

According to the editor, there are no replies needed for this review

Reviewer 3 Report

Review

The manuscript titled "Periurban transformations in the Global South and their impact on water-based livelihoods" by Butsh and Henkel assessed based livelihoods in periurban areas of the Global South. Transformations of water-based livelihoods in periurban areas were also evaluated in terms of their sustainability. The work is interesting. Please check all the detailed comments and questions provided below.

Detailed Comments

Materials and Method

Please correct table 1

Discussion

There is no comparison in the discussion with other countries

Regards

Author Response

Response to reviewer 3:

Comment 1: Please correct table 1

Response: Yes, the table was now formatted according to the journal’s guidelines

Comment 2: There is no comparison in the discussion with other countries

Response: We provide an overview of the literature on the Global South, so this covers a large number of countries already. We are a little uncertain, which comparison would then be useful here and would like to either ask the reviewer to maybe rephrase this suggestion, so that we could pick this up and alter the text or just skip this revision. The latter would be our preferred option, as the review is not country specific.